# Macroscopic Synovial Inflammation Correlates with Symptoms and Cartilage Lesions in Patients Undergoing Arthroscopic Partial Meniscectomy: A Clinical Study

**DOI:** 10.3390/jcm11154330

**Published:** 2022-07-26

**Authors:** Eleonora Olivotto, Giovanni Trisolino, Elisa Belluzzi, Antonello Lazzaro, Alessandro Strazzari, Assunta Pozzuoli, Augusto Cigolotti, Pietro Ruggieri, Andrea Evangelista, Francesca Ometto, Stefano Stallone, Steven R. Goldring, Mary B. Goldring, Roberta Ramonda, Brunella Grigolo, Marta Favero

**Affiliations:** 1RAMSES Laboratory, RIT Department, IRCCS Istituto Ortopedico Rizzoli, 40136 Bologna, Italy; brunella.grigolo@ior.it; 2Reconstructive Hip and Knee Joint Surgery, IRCCS Istituto Ortopedico Rizzoli, 40136 Bologna, Italy; giovanni.trisolino@ior.it (G.T.); alestrazzari@gmail.com (A.S.); 3Pediatric Orthopedic and Traumatology, IRCCS Istituto Ortopedico Rizzoli, 40136 Bologna, Italy; stallone.stefano@gmail.com; 4Musculoskeletal Pathology and Oncology Laboratory, Department of Surgery, Oncology and Gastroenterology, University of Padova, 35128 Padova, Italy; elisa.belluzzi@unipd.it (E.B.); assunta.pozzuoli@unipd.it (A.P.); 5Orthopedics and Orthopedic Oncology, Department of Surgery, Oncology and Gastroenterology, University-Hospital of Padova, 35128 Padova, Italy; antonello.lazzaro@gmail.com (A.L.); augusto.cigolotti@aopd.veneto.it (A.C.); pietro.ruggieri@unipd.it (P.R.); 6General Affairs Unit, IRCCS Istituto Ortopedico Rizzoli, 40136 Bologna, Italy; a.evangelist@gmail.com; 7Rheumatology Unit, Department of Medicine, University-Hospital of Padova, 35128 Padova, Italy; francesca.ometto@unipd.it (F.O.); roberta.ramonda@unipd.it (R.R.); faveromarta@gmail.com (M.F.); 8Hospital for Special Surgery, Weill Cornell Medical College, New York, NY 10021, USA; goldrings@hss.edu (S.R.G.); goldringm@hss.edu (M.B.G.); 9Internal Medicine Unit I, Ca’ Foncello Hospital, 31100 Treviso, Italy

**Keywords:** osteoarthritis, cartilage degradation, meniscal tear, arthroscopic partial meniscectomy, macroscopic score, synovial inflammation, post-operative outcomes

## Abstract

Background: The aim of the study was to examine the relationship among patients’ characteristics, intraoperative pathology and pre/post-operative symptoms in a cohort of patients undergoing arthroscopic partial meniscectomy for symptomatic meniscal tears. Methods: Clinical data were collected (age, sex, body mass index, time to surgery, trauma). Intraoperative cartilage pathology was assessed with Outerbridge score. Meniscal tears were graded with the ISAKOS classification. Synovial inflammation was scored using the Macro-score. Patient symptoms were assessed pre/post-operatively using the KOOS instrument. Results: In the series of 109 patients (median age 47 years), 50% of the meniscal tears were traumatic; 85% of patients showed mild to moderate synovitis; 52 (47.7%) patients had multiple cartilage defects and 31 (28.4%) exhibited a single focal chondral lesion. Outerbridge scores significantly correlated with patient age, BMI and synovial inflammation. There was a correlation between severity of chondral pathology and high-grade synovial hyperplasia. Pre-operative KOOS correlated with BMI, meniscal degenerative changes and symptom duration. Obesity, time to surgery, presence of high-grade synovial hyperplasia and high-grade cartilage lesions were independent predictors of worse post-operative pain and function. Conclusion: We demonstrated that pre-operative symptoms and post-operative outcomes correlate with synovitis severity and cartilage pathology, particularly in old and obese patients that underwent arthroscopic partial meniscectomy. Importantly, patients with a degenerative meniscal pattern and with longer time to surgery experienced more severe cartilage damage and, consequentially, pain and dysfunction. These findings are fundamental to identify patients suitable for earlier interventions.

## 1. Introduction

The meniscus of the knee is a specialized tissue that plays a role in load transmission, shock absorption, and joint stability [1]. Incidental meniscal damage is a frequent finding in the middle-aged population, even in the absence of overt traumatic injury, and these changes increase with age. Although not all of the individuals with meniscal pathology experience clinically significant knee symptoms, there is evidence that the presence of knee meniscal pathology plays an active role in the pathogenesis and progression of osteoarthritis (OA) [2,3]. OA, therefore, is no longer considered a disease involving only the cartilage, but it is recognized as a disorder that affects all components of the joint [4].

The association between meniscal damage and synovial effusion has been noted by MRI [5]. In patients with traumatic meniscal injury, the synovium retrieved during arthroscopic partial meniscectomy (APM) is frequently inflamed, and inflammation scores are associated with increased pain and dysfunction and a unique chemokine profile [6]. Additional studies have shown that synovial inflammation correlates with joint pain and dysfunction and, importantly, is a major risk factor for more rapid progression of structural joint deterioration in OA [7]. Injured and/or degenerated menisci synthesize and release a spectrum of catabolic enzymes and pro-inflammatory mediators. In addition, meniscal tissue samples from patients with traumatic meniscal injury demonstrate pathological alterations characteristic of tissue from older patients undergoing total knee replacement, suggesting that they have high susceptibility to develop OA [8]. In vitro studies have shown that the addition of conditioned media from cultured meniscal tissues from patients with OA increases the production of inflammatory mediators by cultured synoviocytes [9]. Moreover, a co-culture study showed that inflammatory molecules produced by synovium and meniscus could trigger inflammatory signals in early OA patients [10]. These findings support the concept that there is an important crosstalk between the meniscal tissues and synovium and that this interaction may play an important role in the pathophysiology of OA.

Patients undergoing APM represent a unique opportunity to study the relationship between meniscal, cartilage and synovial pathology, especially in the early clinical stages of OA [11]. Moreover, partial meniscectomy is performed by arthroscopy, which permits direct visual characterization of meniscal pathology, cartilage defects, and synovial inflammation [12]—symptoms not detectable by standard radiographic examination [13].

The aim of the study was to investigate the relationship between synovial inflammation and meniscal structural changes in a cohort of patients undergoing APM for symptomatic meniscal tears. We aimed to identify pre- and intra-operative factors that could predict worse post-operative outcomes.

## 2. Materials and Methods

### 2.1. Study Sample

Consecutive patients undergoing knee APM for symptomatic meniscal tears were recruited within the framework of a multicenter prospective cohort study funded by the Italian Ministry of Health (Project code: GR-2010-2317593). The study was conducted in accordance with the Declaration of Helsinki, and the protocol was approved by the Ethics Committee of the IRCCS Istituto Ortopedico Rizzoli and Padova Hospital (Prot. gen. 0020436 and Prot. 2807P). Patients were enrolled after providing written informed consent and if eligibility criteria were met. The study obtained extensive records of preclinical, intraoperative, and post-operative data from patients undergoing APM for degenerative or traumatic meniscal tears. Enrollment of 135 patients undergoing APM for symptomatic meniscal tears occurred from June 2013 to July 2016 [12]. Four orthopedic surgeons with more than 10 years of experience in knee arthroscopy carried out all of the procedures [12]. Inclusion criteria: meniscal tear identified on preoperative MRI confirmed during arthroscopy and considered to be the cause of symptoms; adult patients (age > 18 years); and ability to provide informed consent for participation. All inclusion criteria needed to be satisfied for patient enrollment. Exclusion criteria were: malignancies and overall poor general condition of health; presence of coagulation disorders; presence of tumors, infections, rheumatic or metabolic diseases or other conditions involving the knee joint, previous history of OA. In addition, patients with history of previous surgery on the affected knee or showing symptoms that suggested systemic inflammatory arthritis (multiple joint complaints, concurrent back pain, etc.) were also excluded in order to limit potential biases.

Moreover, we excluded from the present series those patients (18 cases) in which additional procedures were performed during the arthroscopy other than meniscectomy and those patients (8 cases) with incomplete or missing intraoperative data. Finally, 109 patients were included in the present study. A flowchart illustrating the patients’ selection is shown in Appendix A.

### 2.2. Clinical Data

The following demographic and clinical data were collected at baseline: age, sex, body mass index (BMI), date of injury, time to surgery, history of trauma, and symptom duration. Meniscal tears were categorized as traumatic, for those patients who reported a history of violent trauma and acute onset of symptoms; or degenerative, for those patients who reported no traumatic event or only mild accidents and a slow progression of symptoms over time. All patients underwent a standardized physical therapy program after surgery. The preoperative Knee injury and Osteoarthritis Outcome Score (KOOS) was collected at baseline, 3 months, 1 year and 2 years after surgery [14]. KOOS is a questionnaire used for knee injury that can result in post-traumatic OA as meniscus injury. KOOS consists of the following five subscales: (1) symptoms, (2) pain, (3) function in daily living (ADL), (4) function in sport and recreation (Sport/Rec), and (5) knee-related quality of life (QOL). Standardized answer options are given (five Likert boxes), and each question is assigned a score from 0 to 4. The KOOS instrument was used to evaluate post-operative outcomes. KOOS follow-up questionnaires were administered by investigators blinded for intraoperative findings in order to reduce bias.

### 2.3. Surgical Procedure and Intraoperative Scoring

Meniscal repair or meniscectomy was performed, based on the pathological characteristics of the lesion, following the previously published detailed procedures [12]. After the surgical procedure, the surgeon completed a standardized arthroscopic data collection form as previously described [12]. The data collection form incorporates the International Society of Arthroscopy, Knee Surgery and Orthopaedic Sports Medicine (ISAKOS) classification for meniscal tears [15]; the Outerbridge classification for chondral lesions [16]; and the Macro-score for synovial inflammation performed at four sites (suprapatellar pouch, medial and lateral gutters, and peri-anterior cruciate ligament) [17].

Meniscal tears were graded according to the ISAKOS classification, which included the following characteristics: whether the tear was partial (less than 50% of the meniscus thickness) or complete; the rim width (Zone 1 tears have a rim width of 3 mm; Zone 2 tears have a rim width of 3 to 5 mm; Zone 3 tears have a rim width of 5 mm); the location of the tear using both the anterior, middle, and posterior and anterior posterior classification schemes; tear pattern; quality of the tissue; the “total meniscal score”, calculated as the sum of the areas partially (1 point) or totally (2 points) involved by the lesion (range 0–18 for each meniscus). We did not include in the description the item “central to the popliteal hiatus”, as this refers only to the lateral meniscus.

Cartilage lesions were evaluated at different sites: femoral, tibial and patellar sites graded from 0 to 4 (grade 0: normal cartilage; grade 1: cartilage with softening and swelling; grade 2: a partial-thickness defect with fissures on the surface that do not reach subchondral bone or exceed 1.5 cm in diameter; grade 3: fissuring to the level of subchondral bone in an area with a diameter of more than 1.5 cm; grade 4: exposed subchondral bone). A maximal cartilage grade was calculated considering the highest score assigned in the entire joint. Likewise, patients were subgrouped as low-grade (Outerbridge 1–2) or high-grade (Outerbridge 3–4) cartilage lesions.

The presence of synovial inflammation was scored at four different sites: suprapatellar pouch, medial and lateral gutters, and peri-anterior cruciate ligament (ACL), as already described by Trisolino et al. [12]. The following three characteristics of synovial inflammation were graded on a scale between 0 and 4: hypertrophy, vascularity, and synovitis [17]. A “subtotal synovial score” for each compartment was calculated as the sum of the score of the three parameters (range 0–12), and was categorized as mild (≤3), mild-moderate (4–6), moderate (7–9), and severe (≥10). For statistical analysis, each component of the Macro-score for synovial inflammation was converted to dichotomous variables, defining grade 1–2 as low-grade and grade 3–4 as high-grade.

### 2.4. Sample Size Estimation

G*Power software [18] was used to perform an a priori power analysis to ascertain the required sample size in order to achieve adequate power. The sample size was estimated considering the KOOS achieved at two years of follow-up as outcome measure. Considering at least a small effect size (f2 = 0.1), with 20 predictors potentially affecting the outcome, a power of 80% and a probability of alpha error <5%, 82 cases would be included in the analysis. Moreover, assuming lost to follow-up of 20%, at least 98 participants would be required for this study.

### 2.5. Statistical Analysis

Baseline characteristics of patients are summarised as number (%) for categorical variables and as mean (SD) or median (IQR) for continuous variables. Correlation between meniscal pathology, synovial inflammation component score and severity of chondral damage was evaluated with the Spearman’s or Pearson’s correlation coefficient.

Patient characteristics were also stratified by etiology (traumatic or degenerative), comparing continuous variables with the Mann–Whitney U test and categorical variables with the chi-squared test. Mean differences according to patient characteristics on KOOS subscales during post-surgery follow-up were evaluated using multivariable linear regression models. For each KOOS subscale, we estimated a multivariable model including all of the following variables: gender, age at surgery, BMI, history of trauma, symptom duration (months), site of meniscus lesion (medial, lateral, bilateral), Outerbridge classification (≤2 vs. ≥3), and each component of the Macro-score for suprapatellar synovial inflammation (≤2 vs. ≥3). Effects of considered factors were also adjusted for the baseline level of specific KOOS subscale and time of measurement (3, 12, or 24 months). Due to repeated measurements on the same patient, mean differences were estimated controlling the standard errors using the clustered sandwich estimator [19]. All of the variables were retained in the model, and no automatic variable backward selection procedure was applied since the number of observations per degree of freedom of the models (n = 19) was large enough to prevent overfitting. Analyses were performed using STATA 11.2.

## 3. Results

### 3.1. Patient Demographics and Baseline Characteristics

Patient demographics and clinical features at baseline are summarized in Table 1.

### 3.2. Arthroscopic Findings

#### 3.2.1. Meniscus Pathology

The medial meniscus was affected in 79 patients (72.5%), the lateral meniscus was involved in 14 patients (12.8%), while 16 patients (14.7%) presented tears in both menisci. Patients with both menisci involved had statistically significant worst KOOS at baseline and at follow-up (*p* = 0.01). Meniscal lesions were more frequently located in the posterior horn (43.7%). The most frequent type of injury resulted in the flap tear (34.1%), followed by complex tears (27.3%); bucket handle tears were found in 7.5% of the cases. Intra-operative meniscal tear characteristics are summarized in Table 2.

#### 3.2.2. Cartilage Lesions

Eighty-three patients (76.2%) presented some aspect of cartilage damage, and 52 patients had multiple cartilaginous lesions (62.7%) predominantly localized to the femur and patella. In 38 patients (35.9%), high-grade cartilage lesions (Outerbridge >2) were observed. Intra-operative cartilage characteristics are summarized in Table 3.

#### 3.2.3. Synovial Inflammation

Of the 109 enrolled patients, only 6.42% did not have any signs of synovial inflammation. In general, synovial inflammation was low-grade: mild and mild-moderate (score 3–6) in more than 75% of patients in all compartments analyzed. Only one patient had severe synovial inflammation (score ≥ 10) in the suprapatellar and medial compartment. The suprapatellar compartment was the site most frequently affected by synovial inflammation, confirming data already reported [6]. Indeed, 94 patients (86.2%) showed macroscopic evidence of synovial inflammation at the suprapatellar site scored between mild and moderate on the subtotal score. Only 15 patients (13.8%) did not exhibit any sign of inflammation at this site. The characteristics of synovial inflammation in the suprapatellar compartment are reported in Table 4. Synovial inflammation characteristics at the other sites (medial, lateral and peri-ACL compartments) are reported in Appendix A. Noticeably, there was a slight but significant correlation between suprapatellar synovial hyperplasia with the severity of chondral damage (Spearman rho = 0.357; *p* < 0.0001) (Appendix A). No significant correlations were found comparing synovial membrane characteristics in the other compartments.

Patients were also stratified based on the etiology of meniscal lesions: 57% of patients had meniscal degenerative tears, while 43% had traumatic tears (Appendix A). As expected, most traumatic patients were younger compared to the others with degenerative tears (median age of 43 years versus 51, *p* = 0.002), with extremely shorter symptoms duration before arthroscopy, respectively (0.41 versus 1 median years, *p* < 0.001). Lateral meniscus lesions were more involved in patients with traumatic tears (21% vs. 6%), while bilateral meniscal lesions were more frequent in the degenerative group (23% vs. 4%) (*p* = 0.005). Evaluating the presence of synovial inflammation at different sites, we observed higher synovitis grades in the lateral compartment in degenerative patients (*p* = 0.036). No differences were found between the two groups regarding sex, BMI, cartilage defects, synovial inflammation and baseline KOOS subscales (Appendix A).

### 3.3. Clinical Outcomes

Overall, patients reported an improvement in total and in each domain of the KOOS at three months post-operatively, which was maintained at one and two years of follow-up (Appendix A). Concerning the pre-operative clinical data, obesity (BMI > 30), degenerative meniscal tear, and longer time to surgery were independent predictors of worse pain and symptoms at follow-up (Table 5). Furthermore, older patients and patients with lateral meniscal tears presented lower values in KOOS sport and quality of life (QOL) domains. The presence of high-grade cartilage lesions was associated with poorer post-operative outcomes across all domains of the KOOS (Table 4 and Figure 1). In addition, the presence of high-grade suprapatellar synovial hyperplasia was a strong independent predictor of more severe symptoms, pain, and worse function in daily living (ADL) KOOS subscales at follow-up (Table 5 and Figure 1).

## 4. Discussion

Our study was undertaken to characterize the state of meniscal tissues, synovium and articular cartilage at the time of arthroscopic surgery in a population of patients undergoing APM for symptomatic meniscal tears. The goal was to gain insights into the relationships between meniscal and chondral pathology and the grade of synovial inflammation and pre- and post-operative pain and function. We were interested in identifying clinical and intra-operative factors as predictors of worse post-operative outcomes in order to identify patients suitable for earlier interventions. Our results indicate that suprapatellar synovial hyperplasia and additional clinical variables, including obesity, degenerative tears, and symptom duration, were independent predictors of pain and symptoms at baseline and follow-up. According to the literature [16,17], obese patients had increased pain and symptoms at the time of surgery and in follow-up, probably due to the low-grade inflammation status and mechanical overloading.

As predicted, we found that most of the traumatic patients were younger compared to the others with degenerative tears, showing an extremely shorter symptoms duration before arthroscopy. Older patients without a history of trauma were at higher risk of developing symptoms at follow-up. We attribute this to the presence of a degenerative pattern [20] of meniscal pathology that is believed to be a key feature of the early knee OA [21]. Our findings are in line with previous studies reporting that degenerative meniscal lesions are associated with worse clinical and functional outcomes compared to traumatic meniscal tears [22]. Conversely, this assumption was not confirmed in recent prospective studies comparing the effects of APM for traumatic and degenerative meniscal tears [23].

Concerning the meniscal lesion characteristics, we confirmed that patients with involvement of both menisci showed overall worse clinical features at presentation and post-operatively [21] and that lateral meniscus lesions, mostly present in the degenerative group of patients, had poorer outcomes, especially in sports activities [24]. As previously reported, the presence of high-grade cartilage lesions was associated with poorer outcomes [25] and more rapid progression of knee OA [26].

The role of meniscal tears and cartilage lesions in the onset of pain and symptoms in patients undergoing APM has been extensively investigated [27,28,29]. Conversely, the role of synovial inflammation in pathology due to meniscal tears has not been adequately evaluated so far, with most studies investigating the role of synovial inflammation in patients with high-grade synovitis and end stage [6,30,31]. There is less information on the role of low-grade synovial inflammation in the early stages of knee OA in OA patients undergoing total knee replacement. Previous studies confirm that the Macro-score for synovial inflammation is a reliable and reproducible tool for rating the intraoperative state of synovium [12,13]. In our studies, we confirmed that the suprapatellar pouch is the most common site to detect synovial inflammation during arthroscopic meniscectomy. In addition, we observed that most of the patients showed macroscopic evidence of low-grade synovitis (86.24%) and that suprapatellar synovial hyperplasia was an independent predictor of worse symptoms and pain pre-operatively and at follow-up. Although there is lack of consensus regarding the definition of early knee OA, our findings suggest that patients with meniscal tears and associated joint symptoms might be studied as a relevant model of early knee OA [11,21]. The relationship between synovitis and meniscal tear was evaluated by MRI in few studies. In patients without OA, synovial effusion was detected in 44.9% of knees with meniscal damage vs. 30.6% in those without meniscal damage. The adjusted odds ratio of effusion in a knee with meniscal damage was 1.8 [5]. In another study, extensive effusion-synovitis was found at baseline in 48% of the patients, while minimal in 52% [32]. In addition, the presence of persistently minimal effusion-synovitis was found in 45% of the patients, intermittent in 33%, and persistently extensive in 21% over the 18 months of follow-up. Participants with persistently extensive effusion-synovitis were 79% female and had a mean BMI of 31, and a positive association was found over 18 months between the presence and persistence of extensive effusion-synovitis and progression of cartilage damage depth [32].

Scanzello et al. analyzed 33 patients undergoing APM for symptomatic meniscal tear and observed microscopic suprapatellar synovial inflammation and hyperplasia on surgical biopsies in 43% of the patients [6]. They found a significant association between low-grade synovitis and worse pain and function scores at baseline, although they could not confirm this relationship at post-operative follow-up in the first two years after arthroscopic surgery [6,30]. They concluded that larger cohorts and longer follow-up should be pursued to determine if the presence of synovial inflammation at the time of surgery was predictive of poorer long-term symptomatic and functional outcomes and/or the development of clinically detectable OA.

In our recent histological study, we confirmed Scanzello’s results, showing in patients undergoing arthroscopic partial meniscectomy for meniscal tear that microscopic synovial inflammation was associated with pre-operative total KOOS scores, knee symptoms, and pain [33]. We also demonstrated that in meniscal samples, the expression of metalloproteinase-13 (MMP-13), one of the key effectors in the cartilage degradation network in OA, was associated with pre-operative pain, cartilage degeneration, BMI, and time to surgery.

In the present study we were able to confirm that suprapatellar synovial hyperplasia detected during arthroscopic surgery was associated with worse pre-operative symptoms, [30] and, most importantly, it was an independent predictor of post-operative symptoms, pain, and worse ADL at follow-up. Moreover, synovial hyperplasia was associated with the severity of chondral damage (Figure 1 and Table 5). Additionally, in vitro evidence confirmed the interaction between synovial inflammation and meniscal tear. In our previous study, we demonstrated that addition of conditioned media from cultured meniscal tissues from patients with OA increases the production of inflammatory mediators by cultured synoviocytes [9]. More recently, Ogura et al. compared the levels of tumour necrosis factor-alpha (TNF-α), interleukin-6 (IL-6), and nerve growth factor (NGF) in injured and uninjured meniscal tissue and observed increased levels of these inflammatory cytokines in the injured compared to the normal meniscal tissue [28].

The present study has several limitations. No graduated probes were used during arthroscopy; thus, millimetric evaluation of the lesions was not possible. This could determine an overestimation of the size of the lesions [34], due to the artefact of magnification. Moreover, no information about pain medications was collected at the baseline and thus, it cannot be excluded that this could have influenced the clinical outcome and pain assessment. Finally, the arthroscopic procedures were performed using different instrumentation and set-ups. This is critical in particular for the evaluation of synovial inflammation, because the vascularization and hyperaemia can be affected by fluid pressure and the use of a tourniquet [35].

## 5. Conclusions

The present study, importantly, demonstrated that patients with meniscal tear and synovial hyperplasia, especially if older, obese, with a degenerative meniscal pattern and with longer time to surgery, progressed to more severe cartilage damage and, consequentially, pain and dysfunction. We highlight the importance of identifying these patients because they are suitable for earlier interventions. Consequentially, targeting synovial inflammation in patients undergoing meniscectomy might be an optimal strategy in order to prevent cartilage degradation and reduce pain and dysfunction. In clinical practice, patients with meniscal tear showing risk factors for disease severity and, consequently, worse post-operative outcomes should be treated more aggressively and followed up more closely in order to prevent joint damage. Although our results provide further evidence for a role of meniscal pathology in OA pathogenesis, the underlying pathophysiological mechanisms by which meniscal damage and/or degeneration contribute to the synovial inflammation and cartilage pathology and the role of these changes in the pathogenesis and progression of OA need to be further elucidated.

## Figures and Tables

**Figure 1 jcm-11-04330-f001:**
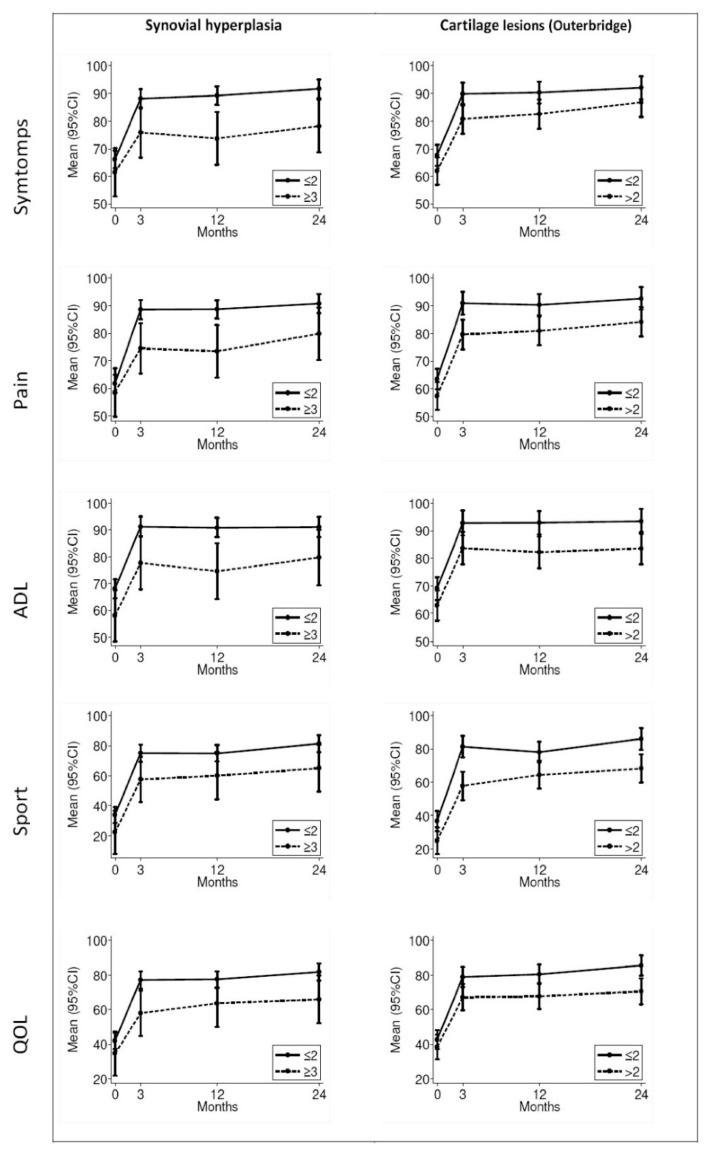
Mean (95% CI) KOOS subscale changes during post-surgery follow-up in patients with high and low grade of suprapatellar synovial hyperplasia (≤2 vs. > 3) and cartilage lesions (≤2 vs. >2) (ADL = Function in daily living; QOL = Knee-related quality of life; CI = confidence interval).

**Table 1 jcm-11-04330-t001:** Patient demographic and clinical characteristics at baseline (N =109).

Men/Women, Number (%)	74 (67.89)/35 (32.11)
Age mean ± (SD), years	47.45 ± (11.03)
BMI mean ± (SD), kg/m^2^	26.94 ± (4.75)
History of trauma, number (%)	47 (43.11)
Right/Left knee, number (%)	56 (51.38)/53 (48.62)
Symptom duration, median (IQR) years	0.77 (1.52–0.32)
KOOS total median (IQR), %	60.71 (72.62–47.62)
KOOS SPT subscale mean ± (SD), %	65.60 ± 20.39
KOOS pain subscale mean ± (SD), %	61.29 ± 19.09
KOOS ADL subscale median (IQR), %	69.12 (85.29–51.47)
KOOS Sport/Rec subscale mean (SD), %	32.43 ± 23.56
KOOS QOL subscale mean ± (SD), %	41.06 ± 18.89

The data are shown as means ± standard deviations (SD) or medians and interquartile ranges (IQR) depending on variable distribution. KOOS = Knee injury and Osteoarthritis Outcome Score (KOOS), SPT= symptoms, ADL = Function in daily living, Sport/Rec = Function in sport and recreation, QOL= Knee-related quality of life.

**Table 2 jcm-11-04330-t002:** Meniscus Tear Characteristics (N = 109).

Location (n. Total Meniscal Lesions 128/128)	Medial Meniscus Tear, Number (%)	79 (72.48)
	Lateral meniscus tear, number (%)	14 (12.84)
	Bilateral meniscus tear, number (%)	16 (14.68)
	Posterior horn, number (%)	56 (43.75)
	Posterior horn-body, number (%)	33 (25.78)
	Body, number (%)	16 (12.50)
	Anterior horn-body, number (%)	8 (6.25)
	Anterior horn, number (%)	7 (5.47)
	Entire meniscus, number (%)	8 (6.25)
Thickness (n. 126/128)	Partial/Full thickness, number (%)	106 (84.13)/22 (17.46)
Type (n. 126/128)	Radial, number (%)	13 (10.32)
	Longitudinal-vertical, number (%)	21 (16.67)
	Horizontal, number (%)	4 (3.17)
	Flap, number (%)	43 (34.13)
	Bucket-handled displaced, number (%)	10 (7.49)
	Complex, number (%)	35 (27.28)
Length (n. 122/128)	0–9 mm, number (%)	24 (19.67)
	10–19 mm, number (%)	65 (51.28)
	>20 mm, number (%)	33 (27.05)

**Table 3 jcm-11-04330-t003:** Cartilage lesion features (N = 109).

Presence/Absence of Cartilage Lesions, Number (%)	83 (76.15)/26 (23.85)
Single/Multiple, number (%)	31 (37.34)/52 (62.65)
Femoral lesions, number (%)	70 (46.5)
Medial condyle, number (%)	56 (53.33)
Lateral condyle, number (%)	18 (17.14)
Trochlear, number (%)	31 (29.52)
Tibial lesions, number (%)	32 (21.05)
Medial tibial plateau, number (%)	31 (63.27)
Lateral tibial plateau, number (%)	18 (36.73)
Patellar lesions, number (%)	50 (32.89)
0–9 mm, number (%)	3 (1–6)
10–19 mm, number (%)	32 (21.05)
>20 mm, number (%)	31 (63.27)
Maximal Cartilage grade, number (107/109)	
0	26 (24.30)
1	10 (9.3)
2	33 (30.84)
3	25 (23.36)
4	13 (12.5)

**Table 4 jcm-11-04330-t004:** Synovial inflammation characteristics (0–4) and subtotal synovial score in the suprapatellar pouch (0–12) (N = 109).

Synovial Inflammation Characteristics, Number (%)						
		0	1	2	3	4
Hypertrophy		24 (22)	44 (40)	28 (26)	11 (10)	2 (2)
Vascularity		36 (33)	29 (27)	32 (29)	11 (10)	1 (1)
Synovitis		29 (27)	39 (36)	28 (26)	11 (10)	2 (2)
**Subtotal Synovial Score, Number (%)**						
		**Absent** **0**	**Mild** **(≤3)**	**Mild-Moderate** **(4–6)**	**Moderate** **(7–9)**	**Severe** **(≥10)**
		15 (13.76)	44 (40.36)	36 (33.03)	13 (11.93)	1 (0.92)

**Table 5 jcm-11-04330-t005:** Multivariable regression models *.

	Symp			Pain			ADL			Sport			QOL		
	Coef	95%CI	*p*	Coef	95%CI	*p*	Coef	95%CI	*p*	Coef	95%CI	*p*	Coef	95%CI	*p*
**Age (1-y inc)**	−0.05	−0.25, 0.14	0.592	−0.12	−0.31, 0.06	0.189	−0.14	−0.34, 0.06	0.172	−0.46	−0.84, −0.08	0.019	−0.33	−0.71, 0.05	0.086
**MALE (1)**	−0.51	−4.89, 3.87	0.818	−0.92	−5.38, 3.53	0.682	−0.04	−4.85, 4.76	0.985	−1.16	−9.77, 7.44	0.789	−4.03	−12.05, 3.99	0.322
**BMI 25–30 vs. <25**	3.23	−1.54, 8.00	0.182	4.3	−0.37, 8.97	0.071	3.32	−1.41, 8.06	0.167	2.09	−6.59, 10.77	0.634	4.76	−3.57, 13.10	0.26
**BMI >30 vs. <25**	4.86	0.08, 9.64	**0.046**	5.65	0.48, 10.82	**0.033**	3.86	−1.65, 9.38	0.168	5.61	−3.19, 14.41	0.209	5.14	−3.78, 14.05	0.256
**Degenerative MT**	4.52	0.31, 8.74	**0.036**	2.6	−1.62, 6.82	0.225	4.06	−0.15, 8.27	0.058	6.01	−2.02, 14.05	0.141	0.18	−7.99, 8.34	0.966
**Time to S (1-*p* inc)**	−1.33	−2.34, −0.32	**0.011**	−1.37	−2.65, −0.09	**0.036**	−1.08	−2.38, 0.21	0.101	−1.43	−3.31, 0.44	0.133	−2.27	−4.10, −0.45	**0.015**
**LM vs. MM**	−5.35	−11.19, 0.49	0.072	−3.43	−8.41, 1.56	0.176	−3.69	−8.50, 1.12	0.131	−10.56	−19.09, −2.03	**0.016**	−8.85	−17.28, −0.42	**0.04**
**Bilateral vs. MM**	−2.1	−7.52, 3.33	0.445	−1.44	−6.54, 3.65	0.575	−0.48	−5.41, 4.44	0.846	−6.14	−16.08, 3.80	0.224	−0.94	−13.68, 11.79	0.883
**Cart deg G tot > 2**	−5.64	−11.27, −0.01	**0.05**	−7.55	−12.95, −2.16	**0.006**	−6.89	−12.49, −1.29	**0.016**	−11.58	−20.83, −2.34	**0.015**	−12.4	−22.02, −2.78	**0.012**
**SUPRA H ≥ 3**	−11.94	−17.93, −5.95	**<0.001**	−13.95	−21.71, −6.19	**0.001**	−14.1	−21.89, −6.30	**0.001**	−6.53	−20.50, 7.43	0.355	−7.06	−22.66, 8.54	0.372
**SUPRA V ≥ 3**	0.21	−5.96, 6.38	0.946	−0.38	−9.41, 8.65	0.934	3.56	−4.33, 11.44	0.373	−3.16	−23.92, 17.61	0.764	−7.56	−32.23, 17.12	0.545
**SUPRA S ≥ 3**	1.61	−4.79, 8.02	0.618	5.74	−0.99, 12.48	0.094	4.56	−3.21, 12.33	0.248	3.92	−12.74, 20.58	0.642	2.98	−17.42, 23.39	0.772

* Effect adjusted also for the baseline level of specific KOOS score and month of measurement. Due to repeated measurements on the same patient, effect was estimated controlling the standard errors with the Huber–White Sandwich Estimator. Symp = symptoms; ADL = Function in daily living; Sport/Rec = Function in sport and recreation; QOL = Knee-related quality of life; Coef = coefficient; CI = confidence interval; Cart deg grade tot = total cartilage grade, SUPRA H = suprapatellar hyperplasia, SUPRA V = suprapatellar vascularity and SUPRA S = suprapatellar synovitis; 1-y inc = 1-year increase; MT = meniscal tear; time to S = time to surgery; 1-p inc = (1-point increase); LM = lateral meniscus; MM = medial meniscus; G = grade.

## Data Availability

The data that support the findings of this study are available from the corresponding author, [E.O.], upon reasonable request.

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
