# Peer review of "Macroscopic Synovial Inflammation Correlates with Symptoms and Cartilage Lesions in Patients Undergoing Arthroscopic Partial Meniscectomy: A Clinical Study"

_jcm, 2022, doi:10.3390/jcm11154330_

Round 1
Reviewer 1 Report
This is a well written manuscript, however I do have some inquiries and suggestions:
Methods:
Was this a consecutive series of patients?
How many surgeons?
How was the post-operative protocol standardized? What was the follow up regarding physical therapy and other post-operative modalities?
Results:
Was medication use measured in patients or is that data available
Were additional demographics measured?
Discussion:
What is the clinical utility and applicability of this knowledge one could introduce into practice?
Author Response
RESPONSE TO REVIEWER 1 COMMENTS
This is a well written manuscript, however I do have some inquiries and suggestions:
"Thank you very much for the positive comment and for your suggestions which have helped us to improve our manuscript."
POINT 1: Methods:
Was this a consecutive series of patients?
RESPONSE 1: Yes, thank you for the suggestion. We added the information in the text. Please see, page 2 line 85.
POINT 2: How many surgeons?
RESPONSE 2: Our study involved two Orthopedic Units: the IRCCS Istituto Ortopedico Rizzoli and the Orthopedics and Orthopedic Oncology University-Hospital of Padova. In total four surgeons performed the arthroscopic meniscectomy, two for each unit. We have added this information in the paper (Page 2 Line 94-95). We described in the details the surgical procedure in the following our previous paper cited at the reference 12 in the present manuscript: “Trisolino G, Favero M, Lazzaro A, Martucci E, Strazzari A, Belluzzi E, Goldring SR, Goldring MB, Punzi L, Grigolo B, Olivotto E. Is arthroscopic videotape a reliable tool for describing early joint tissue pathology of the knee? Knee. 2017”.
POINT 3: How was the post-operative protocol standardized? What was the follow up regarding physical therapy and other post-operative modalities?
RESPONSE 3: We thank the reviewer for this important question. All patients received the following standardized post-operative protocol:
0-14 days
- Diminish pain, swelling (Ice, Rest, avoid excessive loading and prolonged sitting, use elastic bandage, acetaminophen)
- Wound medication
- Gait training with crutches and partial weight bearing
- Knee exercises (passive knee extension, quad sets, patella mobilization straight leg raise, butt kicks
15-30 days
- Progressive full weight bearing
- Closed chain strengthening exercises
- Proprioception training
- muscle re-education,
- stationary bicycle,
- swimming as tolerated.
After one month the patients were allowed for gradual return to work, sportive and recreational activities.
We have added the following sentence in the manuscript (Page 3 Line 116-7): “All patients underwent a standardized physical therapy program after surgery”.
POINT 4: Results:
Was medication use measured in patients or is that data available
RESPONSE 4: We thank the reviewer for this comment. We are sorry but this information was not collected at the time of the surgery. It is possible that pain medications as NSAIDs might have influenced the grade of inflammation in the joint tissues as well as the patient reported outcomes. We added the following sentence in the limitation session of the discussion: “Moreover, no information about pain medications was collected at the baseline or measured at follow-up, thus, it cannot be excluded that this could have influenced clinical outcome and pain assessment.” (Page 11, Line 349-51).
POINT 5: Were additional demographics measured?
RESPONSE 5: Unfortunately, we have collected only the demographic data described in the manuscript: age, sex and body mass index.
POINT 6: Discussion:
What is the clinical utility and applicability of this knowledge one could introduce into practice?
RESPONSE 6: As already reported in the manuscript, our study was undertaken to characterize the state of meniscal tissues, synovium and articular cartilage at the time of surgery in patients undergoing arthroscopic partial meniscectomy for symptomatic meniscal tears. The goal was to gain insights into the relationship between meniscal and chondral pathology and the grade of synovial inflammation and pre- and post-operative pain and function to identify clinical and intra-operative factors predictors of worse post-operative outcomes. The present study, importantly, demonstrated that patients with meniscal tear and synovial hyperplasia, especially if older, obese, with a degenerative meniscal pattern and with longer time to surgery, underwent to more severe cartilage damage and, consequentially, pain and dysfunction. That information might help the to identify patients suitable of earlier interventions. Consequentially, targeting synovial inflammation in patients undergoing meniscectomy might be an optimal strategy in order to prevent cartilage degradation and reduce pain and dysfunction.
We have added in the text the following sentence in the conclusion at the Page 11 Line 362-65: “In the clinical practice, patients with meniscal tear showing risk factors for disease severity and, consequently, worse post-operative outcomes should be treated more aggressively and followed up more closely in order to prevent joint damage”.
A possible implication is the creation of comprehensive systemic scores of early knee OA, which consider synovial inflammation, along with the status of meniscus and cartilage, in order to stratify more accurately the risk of OA progression and establish early intervention.
Reviewer 2 Report
This is an article addressing the association between synovial inflammation and meniscu tear .This writing and structure was decent .My question was as follows:
As we know the synovial inflammation was also an important result of osteoarthritis , it seemed that you didn't consider the grade of OA as an confounding factor , which may lead to the mis-interpretating the result.
Author Response
RESPONSE TO REVIEWER 2 COMMENTS
This is an article addressing the association between synovial inflammation and meniscus tear. This writing and structure was decent. My question was as follows:
POINT 1: As we know the synovial inflammation was also an important result of osteoarthritis, it seemed that you didn't consider the grade of OA as an confounding factor, which may lead to the mis-interpretating the result.
RESPONSE 1: We thank the reviewer for raising this issue, which allow us to clarify how we assessed the presence of cartilage lesions. As matter of fact, we didn’t use a radiograph scoring to establish the grade of OA as well as Kellgren-Lawrance, but we assessed the presence of cartilage defects by Outerbridge score performed during the arthroscopic meniscectomy by experienced surgeons. Radiographic scores are useful to grade OA when we are studying late OA patients, since x-rays are able to identify only osteophytes, joint space narrowing and sclerosis, but not joint soft tissues. On the contrary, when we are studying early OA patients as well as patients undergoing meniscectomy for meniscal tear, radiograph might not show changes, while arthroscopy is directly able to visualized and quantify in details the presence of cartilage lesions (site, size and grade; please see the Table 3), other than synovial inflammation and meniscal degeneration.
We disagree with the reviewer regarding the fact that we didn’t consider the grade of OA as a confounding factor. As matter of fact, cartilage degeneration was included as a possible confounding factor in the regression model performed to evaluate the predictors of worse clinical outcome after meniscectomy, as you can clearly see at the Table 5.
Reviewer 3 Report
The article is of very good quality and the methodology is very appropriate.
However, there are some small errors to be corrected
· P4 Table 1: S for symptom duration
· P5 Table 1: for KOOS Sport/Rc subscale mean (IQR), the representation of the results requires, as for the other IQRs, two values with brackets
· P6 Table 4: there is a shift of the number zero in the line
· P7 Line 234 : Replace “artroscopy” by “arthroscopy”
· P7 Line 238 : A space too many
· P9 Line 296 : Replace “adeguately” by “adequately”
· P9 Line 312 : Replace “dected” by “detected”
· P10 Line 336 : Replace “In the present studywe” by “In the present study we”
· P12 Line 429: There is an extra DOI?
Author Response
RESPONSE TO REVIEWER 3 COMMENTS
The article is of very good quality and the methodology is very appropriate.
"We thank the reviewer for the positive comments and for identifying errors/typos."
POINT 1: However, there are some small errors to be corrected
RESPONSE 1: We have corrected AS FOLLOWS
- P4 Table 1: S for symptom duration
We have corrected that.
- P5 Table 1: for KOOS Sport/Rc subscale mean (IQR), the representation of the results requires, as for the other IQRs, two values with brackets
We apologized for the mistake. We have corrected that as follow: “KOOS Sport/Rc subscale mean (SD)”.
P6 Table 4: there is a shift of the number zero in the line
- P7 Line 234: Replace “artroscopy” by “arthroscopy”
- P7 Line 238: A space too many
- P9 Line 296: Replace “adeguately” by “adequately”
- P9 Line 312: Replace “dected” by “detected”
- P10 Line 336: Replace “In the present studywe” by “In the present study we”
We have corrected all the previous typos highlighted. Thank you.
- P12 Line 429: There is an extra DOI?
We have deleted the extra DOI. Thank you.
Round 2
Reviewer 2 Report
Thank you fot authours reply.Although your study seems reasonable , it possess major flaw warrented for resolve.Synovial inflammtion in knee pathology was considered as a chronic status and reult from the initial knee injury .The immune cascade had been provocated and then result in synovial inflammation .Difference from synovitis in RA ,it was a slowly process.
1.The object of this study seem to elucidate the usefulness of arthroscopy that can fully assess the synovial inflammatory part and chondrocyte part and thus can predict the outcome. However , the OA stage of these patients and age of this group was extremity importatnt becasue "whehter these synovial inflammation came from mesical tear " or " whether the synovial inflammation came from presented long-term-OA sequela " .You just mix these two up. The classification of patient groups was warrented otherwise your result was of no meaning.
2. You should provide the regression equation more clearly .Did you put all factor (age ,BMI ..) at the same equation ? You seem just put 1 factor at a time and claimed significant.
3. Good work in arthroscopy observation of cartilage and synovial condtion and there is no doubt that it's the best way to observe , howevere , these major major flaws in your design and statistics render the scientific importance
Author Response
RESPONSE TO REVIEWER 2 COMMENTS
Thank you fot authours reply.Although your study seems reasonable , it possess major flaw warrented for resolve.Synovial inflammtion in knee pathology was considered as a chronic status and reult from the initial knee injury .The immune cascade had been provocated and then result in synovial inflammation .Difference from synovitis in RA ,it was a slowly process.
We thank the reviewer for the comment.
POINT 1. object of this study seem to elucidate the usefulness of arthroscopy that can fully assess the synovial inflammatory part and chondrocyte part and thus can predict the outcome. However , the OA stage of these patients and age of this group was extremity importatnt becasue "whehter these synovial inflammation came from mesical tear " or " whether the synovial inflammation came from presented long-term-OA sequela " .You just mix these two up. The classification of patient groups was warrented otherwise your result was of no meaning.
RESPONSE 1. We thank the reviewer for this comment. Patients with a previous history of OA had been excluded from the study. The mean age of the patients (mean 47.45 ± 11.03 years), the short duration of the symptoms (median 0.77 (1.52-0.32) years) and the exclusion of patients with a previous history of surgical procedures of the knee also ensure that none of the patients had a long-term disease of the knee. We further specified this in the Methods section as follows:” Exclusion criteria were: malignancies and overall poor general condition of health; presence of coagulation disorders; presence of tumors, infections, rheumatic or metabolic diseases or other conditions involving the knee joint, including osteoarthritis.”
It is known that patients with meniscal tear may present as cartilage lesions also if not meeting the criteria for a diagnosis of osteoarthritis (Scanzello CR. Arthritis Rheum. 2011; Favero, Osteoarthritis and Cartilage 20 (2012) S54–S296).
In the study, the extent of cartilage damage was accurately described in all patients and adequately considered in the analysis. In fact, the effect of synovial inflammation was adjusted for the presence of cartilage damage, and still following multivariate analysis it resulted to be associated with pain and post-operative outcomes independently from cartilage damage.
POINT 2. You should provide the regression equation more clearly. Did you put all factor (age, BMI ..) at the same equation ? You seem just put 1 factor at a time and claimed significant.
RESPONSE 2. We thank the reviewer for this question. We better clarified in the “Statistical analysis” section and in the caption of the table 5 that the models were estimated including all the mentioned variables, without any further selection. To simplify the reading of the table, we have not reported the effects of the baseline levels and the period as they are generally mandatory variables to be included in the models for repeated measures, but less relevant for the purposes of this analysis. However, we specified how the models were also adjusted for the baseline value of the scores and the measurement period.
POINT 3. Good work in arthroscopy observation of cartilage and synovial condtion and there is no doubt that it's the best way to observe , howevere , these major major flaws in your design and statistics render the scientific importance
RESPONSE 3. We are pleased that the reviewer recognizes the importance of our data obtained during arthroscopy. The design of the study was extensively explained in the material and methods section, with a clear flowchart illustrating the patient’s selection in Supplementary Figure 1. The inclusion and exclusion criteria were also specified.
Importantly, the statistical analysis is one of the study strengths and it was performed by an experienced statistician (A.E.). Apart from univariate analysis, a great attention was dedicated to multivariate analysis as explained in the methods. A linear regression analysis was performed for each KOOS subscale including gender, age at surgery, BMI, history of trauma, symptom duration, site of meniscus lesion, Outerbridge classification, and each component of the Macro-score for suprapatellar synovial inflammation in the model. Importantly, the effect of these factors was adjusted for the baseline level of the specific KOOS subscale and time of measurement.